# Functional linkage of gene fusions to cancer cell fitness assessed by pharmacological and CRISPR-Cas9 screening

Gabriele Picco[1,9], Elisabeth D. Chen [1,9], Luz Garcia Alonso [2,3], Fiona M. Behan [1,3], Emanuel Gonçalves[1], Graham Bignell[1], Angela Matchan[1], Beiyuan Fu[1], Ruby Banerjee[1], Elizabeth Anderson[1], Adam Butler[1], Cyril H. Benes[4], Ultan McDermott[1,5], David Dow[3,6,7], Francesco Iorio [1,2,3], Euan Stronach[3,6,7], Fengtang Yang [1], Kosuke Yusa[1], Julio Saez-Rodriguez [2,3,8] & Mathew J. Garnett[1,3]

Many gene fusions are reported in tumours and for most their role remains unknown. As fusions are used for diagnostic and prognostic purposes, and are targets for treatment, it is crucial to assess their function in cancer. To systematically investigate the role of fusions in tumour cell fitness, we utilized RNA-sequencing data from 1011 human cancer cell lines to functionally link 8354 fusion events with genomic data, sensitivity to >350 anti-cancer drugs and CRISPR-Cas9 loss-of-fitness effects. Established clinically-relevant fusions were identified. Overall, detection of functional fusions was rare, including those involving cancer driver genes, suggesting that many fusions are dispensable for tumour fitness. Therapeutically actionable fusions involving *RAF1*, *BRD4* and *ROS1* were verified in new histologies. In addition, recurrent *YAP1-MAML2* fusions were identified as activators of Hippo-pathway signaling in multiple cancer types. Our approach discriminates functional fusions, identifying new drivers of carcinogenesis and fusions that could have clinical implications.

[1] Wellcome Sanger Institute, Wellcome Genome Campus, Cambridge CB10 1SA, UK. [2] European Molecular Biology Laboratory, European Bioinformatics Institute, Wellcome Genome Campus, Cambridge CB10 1SD, UK. [3] Open Targets, Wellcome Genome Campus, Cambridge CB10 1SA, UK. [4] Massachusetts General Hospital, 55 Fruit Street, Boston, MA 02114, USA. [5] AstraZeneca, CRUK Cambridge Institute, Cambridge CB2 0RE, UK. [6] Research and Development, GlaxoSmithKline, Stevenage SG1 2NY, UK. [7] Research and Development, GlaxoSmithKline, Collegeville, PA 19426-0989, USA. [8] Institute for Computational Biomedicine, Faculty of Medicine, Bioquant, Heidelberg University, 69120 Heidelberg, Germany. [9]These authors contributed equally: Gabriele Picco, Elisabeth D. Chen. Correspondence and requests for materials should be addressed to M.J.G. (email: mathew.garnett@sanger.ac.uk)

Oncogenic gene fusions occur in solid tumours and hematologic malignancies, and are used for diagnostic purposes, patient risk stratification, and for monitoring of residual disease[1]. Critically, the chimeric protein encoded by fusions may be a tumour-specific target for treatment, resulting in significant clinical benefit for patients[2,3]. Fusions are often associated with a tissue histology, but can occur at a low frequency in multiple histologies. Gene fusion transcripts are composed of two independent genes formed either through structural rearrangements, transcriptional read-through of adjacent genes, or pre-messenger RNA (mRNA) splicing. The exchange of coding or regulatory sequences between genes can result in aberrant functionality of the fusion protein, and deregulation of the partner genes, including overexpression of oncogenes and decreased expression of tumour suppressor genes (TSGs).

Discriminating between fusions that have a role in cancer fitness and those that do not is a major challenge with important clinical implications[4]. Deep sequencing technology together with sensitive fusion detection algorithms have led to a dramatic increase in the number of reported cancer-associated fusions[5]. Most fusion transcripts are likely the indirect consequence of genomic instability or false-positive events due to error-prone fusion calling. Previous studies have focused on the identification of fusions, or have investigated the function of specific gene fusions; for example, in the setting of acute myeloid leukemia (AML)[6]. The functional role of most cancer-associated fusions has not been investigated.

We have generated large-scale genomic and pharmacological datasets for over 1000 human cancer cell lines as part of the Genomics of Drug Sensitivity in Cancer (GDSC) project[7,8]. These datasets, together with CRISPR-Cas9 genetic screening technology, make it now possible to systematically assess the contribution of fusion transcripts to cancer cell fitness. Here, we report a comprehensive functional landscape of fusions using RNA-sequencing (RNA-seq) data for 1011 human cancer cell lines. We investigate the functional relevance of gene fusions using differential gene expression, drug sensitivity to >350 anti-cancer compounds, and whole-genome CRISPR-Cas9 drop-out screens to identify fusions required for cancer cell fitness. To our knowledge, this study is the first large-scale systematic analysis in human cancer models to unveil the largely unexplored functional role of gene fusions.

## Results

**Landscape of fusion transcripts**. To systematically identify gene fusions in diverse cancer types, we first analyzed RNA-seq data to define fusion transcripts in the GDSC cancer cell lines (1034 samples from 1011 unique cell lines) representing 41 cancer types (Fig. 1 and Supplementary Data 1)[9]. RNA-seq data for 587 cell lines was obtained from the Cancer Genome Hub (CGHub) and 447 cell lines were sequenced at the Sanger Institute[10]. Fusion calling algorithms are prone to detecting false positives from sequencing artifacts and alignment ambiguities[11]. To improve the accuracy of fusion transcript calling, we used three different algorithms, deFuse, TopHat-Fusion, and STAR-Fusion, across all samples[12-14] and applied stringent filtering criteria. In total, 10,514 fusion transcripts were called by more than one algorithm and taken forward for this study (Fig. 1b and Supplementary Data 2). Targeted PCR of 406 putative fusion breakpoints indicated a validation rate of 72%. Furthermore, we compared 23 samples with RNA-seq data from both Sanger Institute and CGHub (Supplementary Data 1), and the proportion of fusions transcripts in both data sources for a given cell line was 70%. The presence of a fusion in a cell line, even in instances where multiple transcripts involving the same partner genes were detected,

was defined as a "fusion event". Thus, we identified 8354 gene fusion events from 10,514 fusion transcripts and, because only a small number of fusions were recurrent, a total of 7430 unique fusions (Supplementary Data 2).

Next, we examined the number of fusion events that occurred in different cancer types. Cell lines had a median of six fusion events and 26% of fusion events were predicted to be in frame. Fusion numbers varied by cancer type (Fig. 1c), with osteosarcoma and breast cancer having the most (median of 16 fusion events per cell line), and kidney cancers and B-lymphoblastic leukemia together with three non-cancerous immortalized human cell lines having the lowest number of fusion events (median = 2). The prevalence of fusion events for each cancer type in our cell lines was slightly higher, but significantly correlated with the frequency reported from the analysis of 9624 patient samples ($p < 0.001$, $R^2 = 0.42$, Pearson's correlation; Supplementary Fig. 1a), indicating that cell lines reflect the frequency of fusions in tumours from different tissues[15]. We identified recurrent known oncogenic fusions events, including *BCR-ABL1* ($n = 11$ cell lines), *NPM1-ALK* ($n = 5$), *EWSR1-FLI1* ($n = 24$), and *TMPRSS2-ERG* ($n = 2$). Of note, only 431 of 7430 (6%) fusions were recurrent, while the remaining were detected in only one cell line (Fig. 1d), indicating that most fusions are rare.

Of the fusion events we identified, 11% have been reported in human tumour samples[15]. For 14.2% of the fusion events, at least one of the fused genes is in the COSMIC Cancer Gene Census, representing an enrichment for cancer genes (odds ratio = 1.8 and $p < 0.001$, Fisher's test). TSGs were enriched as 5′ end partner genes (odds ratio = 2.1 and $p < 0.001$, Fisher's test), while oncogenes were enriched as 3′ or 5′ genes (odds ratio = 2 and 1.8, respectively, $p < 0.001$, Fisher's test). An empirical permutation test identified known oncogenic fusions enriched in specific cancer types that are consistent with their pathognomonic nature, such as *ABL1* fusions in chronic myeloid leukemia (false discovery rate (FDR) <1%, $n = 9$), *EWSR1-FLI1* fusions in Ewing's sarcoma (FDR <1%, $n = 24$) and *FGFR3* fusions in bladder cancer (FDR <1%, $n = 3$) (Supplementary Fig. 1b). In summary, using multiple algorithms and stringent criteria we built a comprehensive landscape of fusions in cancer cell lines, most of which occur at a low frequency, and reflect the prevalence and tissue specificity in tumour samples.

**Fusions impact gene expression**. Fusions may result in altered expression of fusion partner genes[15]. To identify genes whose expression is altered when fused, we first aggregated fusion events that had a common gene partner at the 5′ or the 3′ end to increase sample size and statistical power. We then used multiple linear regression (MLR) to link gene expression with the presence of a fused gene, incorporating bias due to copy number alterations and cancer type. In total, we tested 902 genes (5′ genes $n = 611$ and 3′ genes $n = 383$) that involved 3048 fusions. We identified 172 (19%) genes significantly associated with differential expression (5′ genes = 54 (9%) and 3′ genes $n = 118$ (31%)) that encompassed 592 fusions (Fig. 2a). Of the significantly associated genes, 24 (14%) were from the COSMIC cancer gene census (2.5% of the total; Fig. 2a and Supplementary Data 3). As expected, several TSGs such as *TP53*, *APC*, and *KDM6A* were significantly associated with reduced expression ($p < 0.001$, MLR, Supplementary Fig. 1c). In contrast, many oncogenes fused at the 3′ end were overexpressed, including *ALK*, *ERG*, *FL1*, *MYC*, *MLL4*, and *ROS1* ($p < 0.001$, MLR, Supplementary Fig. 1c).

Because most fusions are rare and therefore not suitable for linear regression modeling, we also annotated expression of genes involved in each fusion event ($n = 8354$). We focused on 3′ end genes with exceptionally high expression because overexpression

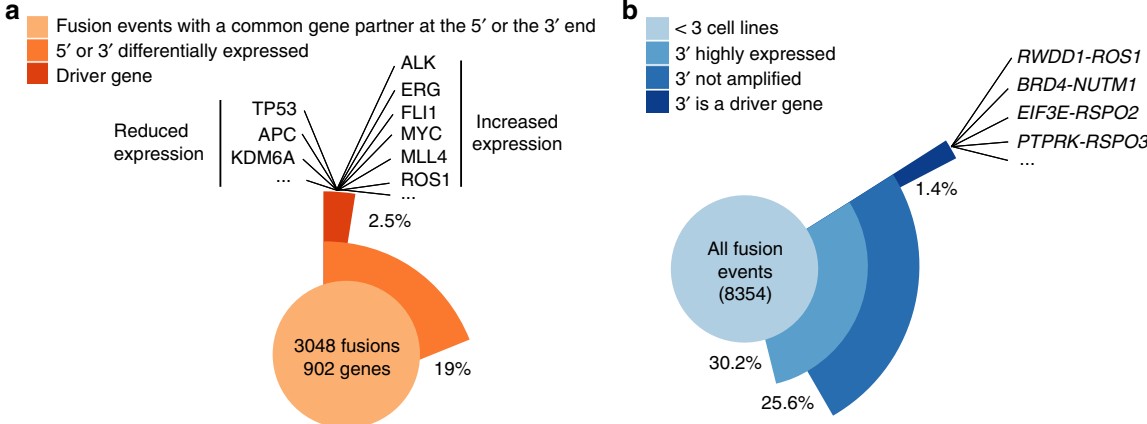

**Fig. 1** Landscape of gene fusions in cancer cell lines. **a** Tissues (inner ring) and cancer types (outer ring) represented by the cell lines and CRISPR dataset used for this study. **b** Fusion transcript calls using three algorithms and their overlap. **c** Frequency of gene fusion events in cancer cell lines, separated by cancer type. The black line is the median. **d** Fusion event recurrence in cancer cell lines

**Fig. 2** Fusions impact gene expression. **a** Frequency of a statistical association between a recurrent fusion (*n* > 2 cell lines) and differential gene expression. Examples of downregulated tumour suppressor genes (TSGs) and overexpressed oncogenes are displayed. **b** Frequency of co-occurrence of a gene fusion and overexpression of the 3′ gene for each fusion event. *RSPO2, RSPO3,* and *NUTM1* are examples of overexpressed cancer driver genes involved in previously unreported gene fusions

of proto-oncogenes occurring as 3′ partner genes is observed in several malignancies[15,16]. We found that 25.6% ($n = 2145$) of fusion events were coincident with high expression and did not co-occur with copy number amplification (Fig. 2b). Only 5.4% (1.4% of the total; $n = 117$) of these fusion events involve the overexpression of a COSMIC driver gene (Fig. 2b and Supplementary Data 4). Thus, aberrant transcript expression of genes involved in gene fusions is a common event, but only a small subset of these fusions involve established driver oncogenes.

This analysis identified new contexts for fusion transcripts leading to overexpression of cancer genes located at the fusion 3′ end, such as *NUTM1*, *RSPO2/3*, and *ROS1* (Fig. 2b). In support of this observation, we validated by Sanger sequencing and fluorescence in situ hybridization (FISH) a previously uncharacterized *RWDD1-ROS1* fusion in the OCUB-M cell line, which is derived from a triple-negative breast cancer (Supplementary Fig. 2a and b). ROS1 is a receptor tyrosine kinase and gene rearrangements leading to ROS1 overexpression are therapeutic biomarkers of response to ROS1 kinase inhibitors in non-small-cell lung cancer and other cancer types (Supplementary Fig. 2c)[17]. The fusion retains the ROS1 protein kinase domain and OCUB-M cells display sensitivity to crizotinib and foretinib, two potent ROS1 inhibitors (Supplementary Fig. 2d and e)[18,19]. Interestingly, in a dataset of 590 breast cancer patients, we identified a triple-negative and a HER2+ tumour-carrying in-frame fusions involving the ROS1 kinase domain[20] (Supplementary Fig. 2f), suggesting that this rare subset of breast cancer patients could be potentially eligible to targeted tyrosine kinase inhibitor-based therapies.

**Fusions as markers of drug sensitivity**. Fusion proteins can impact on clinical responses to therapy. Consequently, we reasoned that differential drug sensitivity in cell lines could be used to identify functional fusions, as well as opportunities for repurposing of existing drugs. We used an established statistical model[8,21] to perform an analysis of variance (ANOVA) linking the 431 recurrent gene fusions ($n \geq 2$ cell lines; in-frame and not in frame fusions) with 308,634 $IC_{50}$ (half-maximal inhibitory concentration) values for 409 anti-cancer drugs (334 unique compounds) screened across 982 cell lines by the GDSC project (Fig. 3a and Supplementary Data 5). The compounds assessed consisted of anti-cancer chemotherapeutics and molecularly targeted agents, including many which are Food and Drug Administration-approved ($n = 46$) or in clinical development ($n = 65$; Fig. 3a). This included data for 155 new compounds and a total of 212,774 previously unpublished $IC_{50}$ values. Preliminary analyses indicated that mutations in cancer driver genes co-occurring with fusions in cell lines were frequent confounders when identifying fusion-specific associations. To control for this, we first identified associations between 717 cancer driver mutations and copy number alterations with drug sensitivity, and then used them as a covariate in the ANOVA to identify fusion-specific associations (Supplementary Data 6). Adding the covariates resulted in 11 fusion associations falling below our threshold for statistical significance. For instance, the association of *NKD1-ADCY7* with BRAF-inhibitor dabrafenib was explained by the presence of a coincident *BRAF* mutation in one highly sensitive cell line (Supplementary Fig. 3a).

We identified 227 large-effect size associations (ANOVA FDR <25% and Glass Delta's >1; the Glass Delta's are a measure of effect size incorporating the standard deviation of the two sub-populations) between gene fusions and drug sensitivity (Fig. 3b; Supplementary Data 7). At the level of individual fusion events, 284 of 1355 (21%) tested fusion events were involved in a significant association with a drug. Most of the strongest

fusion–drug associations were well understood cases, such as sensitivity of *ALK*-fusion-positive cell lines to ALK inhibitors, for example, alectinib, (FDR <0.1%), and sensitivity of *BCR-ABL1* translocation-positive cells to ABL inhibitors, such as imatinib and nilotinib (FDR <0.1%) (Fig. 3b, c). We also identified associations with low-frequency fusions, such as sensitivity to multiple EGFR inhibitors (e.g., cetuximab), in two *CRTC1-MAML2*-fusion-positive cells (FDR <0.1%), mediated as a result of fusion-driven paracrine induction of EGFR signaling[22] (Fig. 3c). Following manual curation, most associations between fusions and drug sensitivity could be readily explained by known interactions ($n = 66$; 30%), mutations in secondary genes ($n = 7$; 3%), and fusions that were either not in frame ($n = 77$; 34%) or not detected in patient samples ($n = 131$; 57%). The remaining associations ($n = 35$; 15%) generally involve poorly described fusions present in two or three cell lines, making drug sensitivities difficult to interpret. This analysis suggests that besides well-established oncogenic fusions, there are few recurrent gene fusions that are associated with differential drug sensitivity, and which could be used as biomarkers for repurposing of existing anti-cancer drugs. We did, however, observe potent sensitivity to specific drugs in individual cell lines with rare fusions.

**Functional fusion analysis using CRISPR-Cas9 loss-of-fitness data**. Our analysis of fusions using drug sensitivity data was limited by their low frequency and the limited number of targets covered by available drugs. Here, we complemented our fusion identification pipeline with CRISPR-Cas9 screens to systematically assess fusion function based on their requirement for cell fitness. To be as comprehensive as possible, we considered all fusions including in-frame and not in frame transcripts. We assembled CRISPR-Cas9 whole-genome drop-out screening data from Project Score at the Sanger Institute[23], Broad Institute DepMap project[24], and Wang et al. [6], which together span 371 cell lines from 33 different cancer types[6,24]. CRISPR loss-of-fitness screens typically target each gene with 5–10 single guide RNAs (sgRNAs) and average their fold changes to calculate gene-level depletion values. By contrast, we took advantage of individual sgRNA fold changes to query the functional importance of gene regions. We mapped the coordinates of the sgRNAs targeting either of the fusion genes, and classified them as mapping or non-mapping sgRNAs, depending on whether they targeted the fusion transcript or not (Fig. 4a). For each gene fusion, we calculated a fusion essentiality score (FES) representing the average difference between the mapping and non-mapping sgRNAs normalized and scaled fold changes (a measure of the cell fitness effect). For transcripts where there were only mapping guides (50% of transcripts), the value of non-mapping guides was set to zero, allowing us to compare the effect in a specific cell line to the average effect across all cell lines. The statistical significance of each FES and FDR were calculated based on 10,000 randomizations of all sgRNA fold changes in the cell lines, and we set a minimum fold change of $-0.45$ between mapping and non-mapping sgRNA (see Methods).

We identified mapping sgRNA for 2821 (26%) fusion transcripts, of which only 99 (4%; representing 103 fusion events) were significantly associated with decreased cell fitness in at least one CRISPR dataset (FES FDR <5%; Fig. 4b and Supplementary Data 8). This corresponded to one or more functional fusions in 55 (16%) tested cell lines. Using gene-set enrichment analysis (GSEA), functional fusions were enriched for transcripts reported in the COSMIC database of oncogenic fusions ($p < 0.001$, GSEA), which included well-known fusions such as *EML4-ALK*, *EWSR1-FLI1*, and *KMT2A-MLLT3* and *TPM3-NTRK1* (Fig. 4b, c and Supplementary Fig. 3b). In addition, amongst the most significant

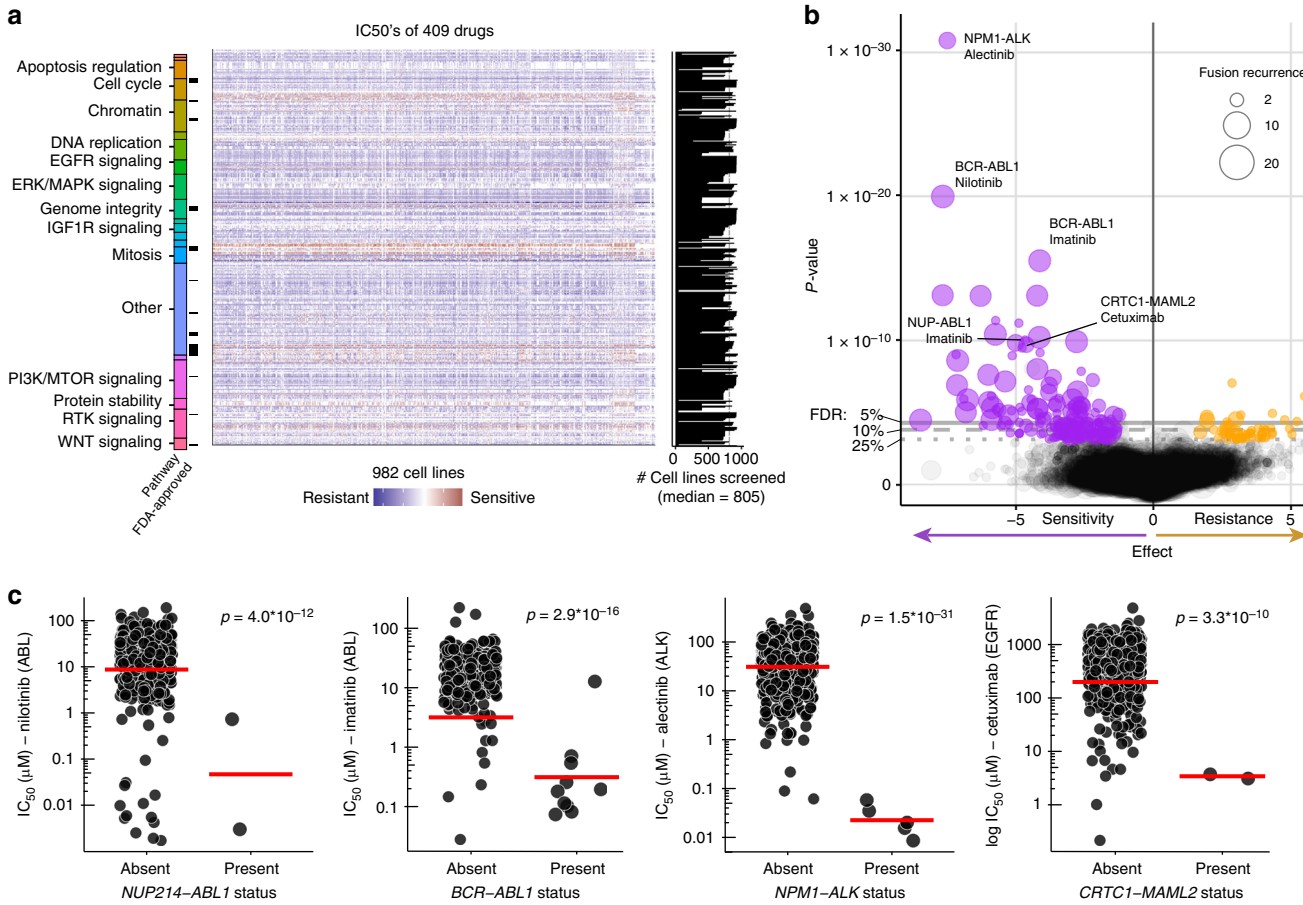

**Fig. 3** Gene fusions as therapeutic biomarkers. **a** Genomics of Drug Sensitivity in Cancer (GDSC) drug sensitivity data utilized (reported as half-maximal inhibitory concentration ($IC_{50}$) values) with Food and Drug Administration (FDA)-approval status of compounds. Compounds are grouped by target or pathway. **b** Analysis of variance (ANOVA) results for fusion–drug associations. Each circle represents a tested association, with circle size indicating the number of cell lines harboring the associated fusion event (fusion recurrence), false discovery rate (FDR) thresholds are indicated. Negative effect sizes are associated with sensitivity and positive effect size resistance. Representative fusion–drug associations are labeled. **c** Examples of differential drug sensitivity in cell lines stratified by fusion status. Nominal therapeutic drug targets are in brackets. Each circle is the $IC_{50}$ for an individual cell line and the red line is the geometric mean. Association significance values (p values) are from the ANOVA test

functional fusions were *YAP1-MAML2* fusions (FDR <0.01%), *DDX6-FOXR1* (FDR <0.5%), and *PICALM-MLLT10* (FDR <0.5%). There was an enrichment in functional fusions for transcripts linked with drug sensitivity based on our ANOVA analysis (p < 0.001, GSEA; Fig. 4b). Interestingly, there was no enrichment for fusion transcripts that were: (i) previously reported in patient samples; (ii) fusion transcripts that are in frame vs. not in frame; (iii) fusion transcripts in amplified regions, which are associated with non-specific fitness effects in CRISPR screens;[25] (iv) nor for fusion transcripts that involve genes in the COSMIC Census[26]. Moreover, when considering only the subset of fusions most likely to be clinically relevant, namely those involving an in-frame event with a COSMIC oncogene, we did not observe an enrichment in functional fusions (Fig. 4b). Overall, for most tested fusions, we did not detect evidence supporting a functional role in cancer cell fitness.

**Function of oncogenic gene fusions across different histologies.** Despite an absence of evidence supporting the function of most fusions, as exemplified in the following analyses, we provide new insights into the pathogenic role of specific gene fusions that point to strategies for repurposing clinically approved drugs in rare subsets of fusion-positive cancers.

Rare *RAF1* fusions occur in patient tumours[27–29] and are biomarkers of response to mitogen-activated protein kinase

pathway inhibition. We identified an in-frame *ATG7-RAF1* fusion in PL18, a pancreatic adenocarcinoma cell line (Fig. 5a). The fusion was confirmed by Sanger sequencing and FISH (Fig. 5b and Supplementary Fig. 4a). The fusion removes the N-terminal regulatory region, but retains an intact *RAF1* protein kinase domain, suggesting it results in constitutive kinase activation. Only mapping sgRNAs targeting the portion of the two genes involved in the fusion were significantly depleted in PL18, resulting in a significant FES (Fig. 5c). Moreover, *ATG7*-fusion-targeting sgRNA were only depleted in PL18 cells and not other pancreatic cell lines (Fig. 5d). Unlike >90% of pancreatic tumours and cell lines that have activating mutations in KRAS[8,30], PL18 has a wild-type *KRAS* allele, but retained potent sensitivity to downstream MEK (mitogen-activated protein kinase kinase) pathway inhibitors trametinib and PD0325901 (Fig. 5e and Supplementary Fig. 4b). An *ATG7-RAF1* rearrangement was previously reported in another *KRAS* wild-type pancreatic cancer model[31]. Furthermore, we mined sequencing data for 126 pancreatic adenocarcinoma patient-derived xenograft (PDX) models and identified an additional *KRAS* wild-type tumour with a *PDZRN3-RAF1* fusion, which conserves the *RAF1* kinase domain (Supplementary Fig. 4f). Together, our analysis supports emerging evidence for rare recurrent and potentially therapeutically actionable *RAF1* rearrangements in *KRAS* wild-type pancreatic cancer.

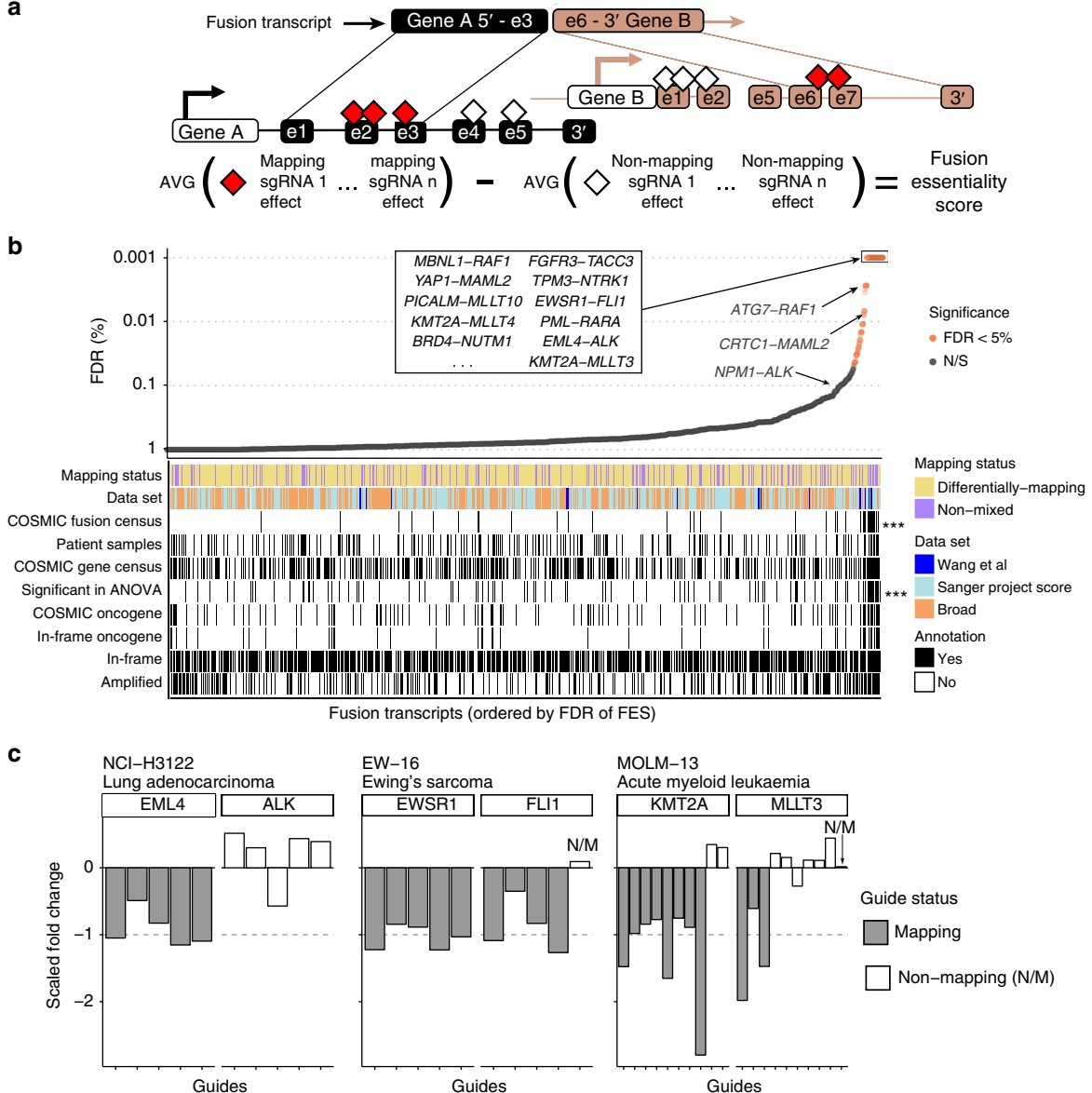

**Fig. 4** CRISPR screening data identifies functional fusions. **a** Calculation of fusion essentiality scores (FES) by measuring the differential fitness effect of mapping vs. non-mapping single guide RNAs (sgRNAs) to each gene in a fusion transcript. **b** False discovery rate (FDR) of FES scores for all testable fusion transcripts ($n = 2821$). Transcripts with at least one mapping and one non-mapping guide are "differentially mapping," while transcripts with only mapping guides are "non-mixed". Fusion transcripts were classified into the indicated genes sets and $p$ values were calculated using gene-set enrichment analysis (GSEA). Selected known oncogenic fusions and other fusions of interest are labeled. **c** Examples of functional fusion transcripts identified in specific cancer cell lines based on FES scoring. Each bar is the scaled fold change of an individual sgRNA to fusion 5′ and 3′ end partner genes, and colored by fusion mapping or non-mapping sgRNA. Dashed line is at −1 (to which known essential guides were scaled). AVG = average; N/S = not significant at 5% FDR

*BRD4-NUTM1* fusions genetically define NUT midline carcinoma (NMC), a rare and aggressive neoplasm that usually arises in the midline of the body with marked sensitivity to BET bromodomain inhibitors[32,33]. We identified a novel in-frame *BRD4-NUTM1* fusion in SBC-3, a cell line established from a 24-year-old male diagnosed with small-cell lung carcinoma (SCLC)[34], and confirmed the fusion by Sanger sequencing and FISH (Fig. 5a, b and Supplementary Fig. 4a). Based on CRISPR data for 206 cell lines screened at Sanger, *NUTM1*-targeting guides were highly depleted only in SBC-3 cells, and the fusion was associated with a significant FES (Fig. 5c and Supplementary Fig. 4c). Moreover, SBC-3 cells displayed marked sensitivity to four different BET inhibitors (Fig. 5e and Supplementary Fig. 4b).

We investigated whether the SBC-3 cell line is mis-classified as an SCLC and is actually a rare NMC of the lung. Unlike >95% of SCLC tumours and cell lines, SBC-3 cells do not have alterations in *RB1* or *TP53*, nor do they express SCLC-specific neuroendocrine markers, such as CgA, NSE, and synaptophysin (Supplementary Fig. 4d). The *BRD4-NUTM1* fusion was specifically associated with high NUTM1 transcript expression in cell lines (Fig. 2b and Supplementary Fig. 4e). Therefore, we mined The Cancer Genome Atlas (TCGA) expression data for SCLC and non-SCLC (NSCLC) searching for samples displaying high *NUTM1* mRNA levels. We identified a single NSCLC sample derived from a 39-year-old patient diagnosed with lung squamous cell carcinoma, displaying *NUTM1* mRNA outlier expression (Supplementary Fig. 4e) and carrying a *NSD3-NUTM1*

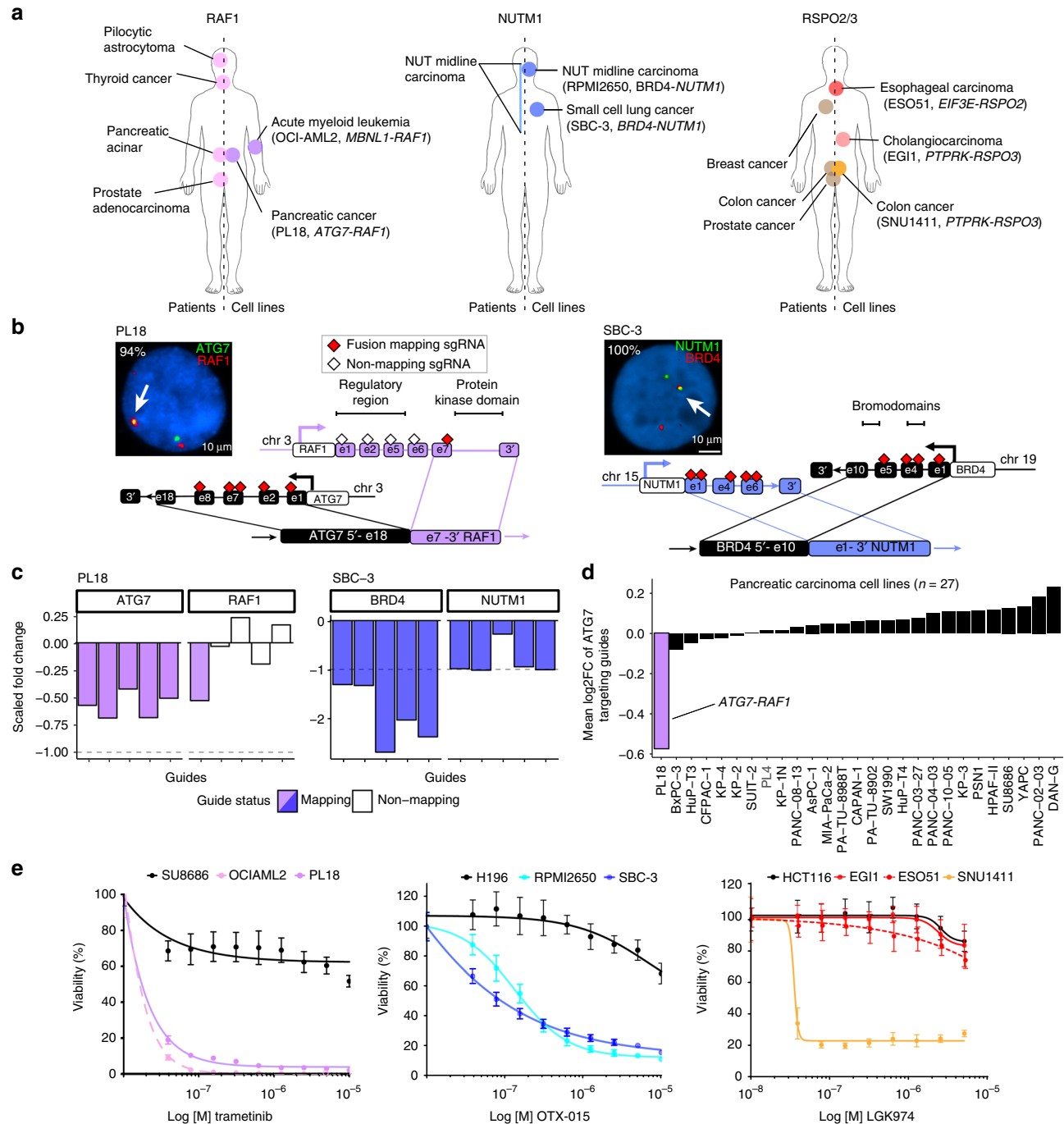

**Fig. 5** Therapeutically actionable oncogenic fusions identified across different histologies. **a** *RAF1, NUTM1,* and *RSPO2/3* fusions identified in patients previously (left) and cell lines in this study (right). Cell lines with known oncogenic fusions used as positive controls are reported. **b** Interphase fluorescence in situ hybridization (FISH) of *ATG7-RAF1* (left) and *BRD4-NUTM1* (right) gene fusions (arrows) in PL18 and SBC-3 cell lines. The percentage of fusion-positive interphases are reported in white text. Schematic representations of each fusions are represented. Only exons involved in the breakpoint or displaying fusion mapping single guide RNA (sgRNAs) (red diamonds) or non-mapping sgRNAs (empty diamonds) are shown. **c** Fold change fusion essentiality score (FES) values of sgRNAs targeting *ATG7* and *RAF1* in PL18 (left) and *BRD4* and *NUTM1* in SBC-3 cells (right). Colored bars indicate values of sgRNAs targeting the exons involved in the fusions. **d** Depletion of fusion-targeting *ATG7* guides for all screened pancreatic cancer cell lines. **e** Viability assay on PL18, SBC-3, EGI-1, and ESO51 cells treated with MEK (trametinib), BET (OTX-015), and PORCN (LGK974) inhibitors, respectively. SU8686, H196, and HCT116 cells are pancreatic, small-cell lung cancer, and colorectal cancer-negative controls. OCIAML2, RPMI2650, and SNU1411 are, respectively, a *RAF1*-rearranged leukemia, a *NUTM1*-rearranged NUT midline carcinoma (NMC), and a *RSPO3*-rearranged CRC cell line included as positive controls. Data are average ± s.d. of three technical replicates and are representative of three independent experiments

rearrangement[15] (Supplementary Fig. 4f), a chimeric oncoprotein recently identified in NMC patients and associated with BET inhibitor sensitivity[35]. *NUT* rearrangements occur in a rare subpopulation of patients diagnosed with SCLC and NSCLC[36,37],

and recent studies established that *NUT*-associated fusions can occur in tumours outside the midline, such as soft tissue, brain, and kidney[38]. Thus, although we cannot exclude misclassification of SBC-3 cells, our preclinical data support evidence that

functional *NUTM1* fusions are present in tumours diagnosed as lung cancer, and could represent an actionable driver event in these tumours with immediate potential clinical implications.

Aberrant expression of RSPO2/3 fusion transcripts synergize with WNT ligands to trigger WNT pathway activation in *APC* wild-type colorectal cancer (CRC)[16]. WNT pathway blockade with porcupine inhibitors is effective in *RSPO3*-rearranged CRC preclinical models[39,40] and clinical trials in patients with *RSPO2/3*-fusion-positive tumours of any histological origin are ongoing (NCT01351103). Here, we detected and validated two unreported canonical R-spondin fusions in cancer cell lines derived from biliary tract (EGI-1; *PTPRK-RSPO3* fusion) and esophagus (ESO51; *EIF3E-RSPO2* fusion) by PCR and FISH (Fig. 5a and Supplementary Fig. 5a, b and c). Aberrant expression of *RSPO2/3* was detected in both cell lines (Fig. 2b and Supplementary Fig. 5d). Similarly, through mining TCGA esophageal cancer data, we found that a tumour with high *RSPO3* expression was positive for a canonical *RSPO3* fusion[15] (Supplementary Fig. 5d). Surprisingly, EGI-1 and ESO51 were insensitive to WNT pathway blockade with a porcupine inhibitor (Fig. 5e and Supplementary Fig. 5e). This was in contrast to SNU1411, a positive control CRC cell line model addicted to WNT pathway activation by rearranged *RSPO3*, which was sensitive to multiple porcupine inhibitors[41]. This result points to an element of tissue specificity in mediating the functional role of some fusions, with potentially important implications for repurposing of WNT pathway inhibitors across different RSPO-fusion-positive tumour types. Interestingly, sgRNA mapping to the *RSPO2/3* fusion were not associated with significant FES in EGI-1 and ESO51, nor in the positive control line SNU1411 (Supplementary Fig. 5f–h). This may be because RSPO2/3 fusions act as paracrine signals to enhance WNT ligands and so are not readily detected by pooled CRISPR screens designed to unveil cell intrinsic gene dependencies.

**Recurrent *YAP1-MAML2* fusions drive Hippo pathway signaling**. We next investigated the function of a poorly understood fusion. Recurrent *YAP1-MAML2* fusions were identified in AM-38 (glioblastoma), ES-2 (ovarian carcinoma), and SAS (head and neck carcinoma) cell lines (Fig. 6a). We validated the fusion in all three cell lines by PCR, and interphase and fiber FISH (Fig. 6b, c and Supplementary Fig. 6a). *YAP1-MAML2* fusions have been reported in nasopharyngeal carcinomas[42] and in a sample from a patient with skin cancer, but not in the three tumour types reported here[15] (Fig. 6a). The fusion brings together exons 1–5 of *YAP1* and exons 2–5 of *MAML2*, a transcript structure that is conserved across all cell lines and patient samples (Fig. 6d and Supplementary Fig. 6b).

A functional role for *YAP1-MAML2* fusions has not been reported. We found that *YAP1-MAML2* fusions were significantly associated with decreased cell fitness when targeted in the CRISPR screen (Figs. 4b, 6e). Furthermore, loss of fitness in response to *MAML2* knockout is unique to *MAML2*-fused cell lines in the three cancer types where the fusion is observed (Fig. 6f). YAP1 aberrant activation is linked with poor prognosis, chemoresistance, and resistance to cell death in multiple solid tumours[43,44]. YAP1 is a transcriptional co-activator of the Hippo pathway through binding with the TEAD1 transcription factor and MAML2 is a transcriptional co-activator involved in NOTCH signaling[45]. *YAP1-MAML2* fuses the transcriptional activation domain of MAML2 with the TEAD-binding domain of YAP1. Intriguingly, ES-2 and AM-38, although not SAS, also showed essentiality for *TEAD1* in the CRISPR drop-out screen (Supplementary Fig. 6d), suggesting that the fusion protein signals through TEAD1.

To further investigate fusion protein activity, we performed GSEA comparing the three *YAP1-MAML2*-fusion-positive cell lines against all others. Of 189 pathways tested, a YAP1-conserved transcriptional signature was the most significant hit (adjusted $p < 0.001$; Fig. 6g). The same signature was highly enriched when ES-2 was compared against all other ovarian cancer cell lines and SAS against all other head and neck cell lines, while expression of prototypic tissue-specific oncogenic signatures, such as estrogen receptor signaling in ovary, were depleted (Supplementary Fig. 6e). Overall, our findings that recurrent *YAP1-MAML2* fusion are associated with increased YAP1 signaling and required for cell fitness support targeting the Hippo signaling cascade in *YAP1-MAML2*-fusion-positive tumours.

**Discussion**
Thousands of gene fusion transcripts have been reported[15,46,47], and most are likely to be passenger events due to chromosomal instability or artifactual. We developed a multi-algorithm fusion calling pipeline, and integrated large-scale genomic and functional datasets, including CRISPR-Cas9 screening data, to systematically identify functional gene fusions across diverse tissue histology. Our analysis is a valuable reference of gene fusions in cancer cell lines. Furthermore, since fusions can have diagnostic, prognostic, and therapeutic utility, our analysis could have clinical implications.

Using our analysis pipeline, we tested 3354 fusion events and found supporting evidence of a functional role for 368 (11.8%) by either CRISPR data ($n = 103$) or drug sensitivity analysis ($n = 284$) (Fig. 6h). Thus, most fusions are likely to be passenger events and dispensable for cell fitness. Of those with functional evidence, only 142 (38.5%) involved a COSMIC cancer gene, 58 (16%) were listed in the COSMIC fusion census, and 107 (29%) were called in TCGA patient samples. Thus, many fusions with supporting functional evidence are poorly understood and do not contain known driver genes, indicating that there are gaps in our knowledge of genes with roles in cancer cell fitness.

Although our analysis suggests that most fusions are not functional, there are several limitations to our approach. Some fusions may be required for aspects of the malignant phenotype not measured here, such as tumour initiation, paracrine signaling, host–tumour cell interaction, and metastasis. Our CRISPR-based approach is only suitable for testing fusions with mapping sgRNA, and is subject to possible bias based on sgRNA efficiency and the number of mapping vs. non-mapping sgRNA. Furthermore, this approach captures fusions that induce gain-of-function or dominant-negative effects, but is not able to identify loss-of-function effects, such as inactivation of a tumour suppressor. Some tumour types are relatively poorly represented, including hematological and pediatric malignancies, and it may be that in some tumour types we observe different patterns of functionality. Finally, sub-clonal fusions could lead to false-negative results. Despite these limitations, our finding that most fusions tested do not have supporting functional evidence, including fusions with cancer drivers genes, emphasizes the importance of analyses to ascribe function when interpreting fusions identified using genomic sequencing.

Gene fusions are used as therapeutic biomarkers to enroll patients in clinical trials and to direct clinical care, often in diverse histologies and clinico-pathologic subtypes. Notable examples are NTRK and ALK fusions, originally identified as effective biomarkers of response to targeted agents in NSCLC patients and occurring at low frequencies (<1%) in a variety of malignancies[48–50]. We provide specific and previously undescribed data on fusions involving *RAF1*, *ROS1*, and *BRD4* that suggest that existing drugs could be repurposed for use in rare pancreatic, breast, and lung cancers. FISH analysis confirmed the presence of these gene fusions in the vast majority of the cell nuclei analyzed (>90%), indicating that these alterations are clonal events. Further studies using tumour xenograft models would support the in vivo efficacy of these findings and could

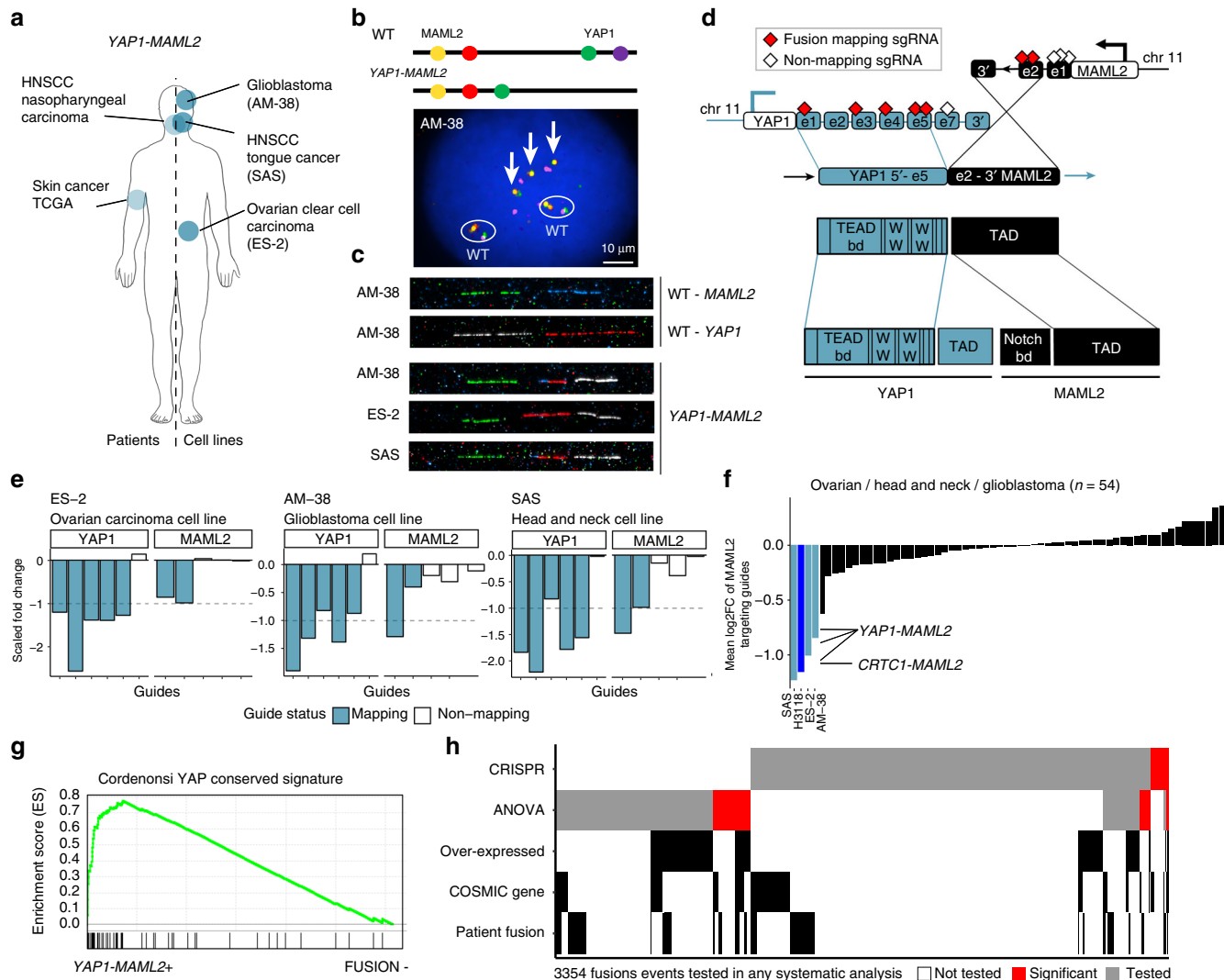

**Fig. 6** Recurrent *YAP1-MAML2* fusions activate Hippo pathway signaling. **a** *YAP1-MAML2* fusions in patient tumours (left) and cell lines (right). **b** Interphase fluorescence in situ hybridization (FISH) (AM-38 only) targeting *YAP1-MAML2* fusion (arrows; cells are polyploid with wild-type chromosomes circled) in AM-38, ES-2, and SAS cell lines. Probes used and chromosomal position are shown schematically. *YAP1* and *MAML2* are both on chromosome 11. **c** Fiber FISH showing gene fusion in AM-38, ES-2, and SAS cell lines. MAML2 probes are shown in green and blue; YAP1 probes are shown in white and red. **d** Schematic of *YAP1-MAML2 and* functional domains involved in the fusion. **e** Fold-change values of single guide RNAs (sgRNAs) targeting *YAP1* and *MAML2* genes in ES-2, AM-38, and SAS cell lines. **f** *YAP1-MAML2* fusion-positive cell lines show the highest depletion of fusion-targeting *MAML2* guides in ovary, head and neck, and glioblastoma cell lines (*n* = 54). Cell line in dark blue (H3118) harbors a known *CRTC1-MAML2* fusion (Fig. 3c)[22]. **g** GSEA of *YAP1* gene signature in *YAP1-MAML2*-positive cells (*n* = 3) vs. fusion-negative (*n* = 1008) cell lines. **h** Heatmap of 3345 fusion events tested in CRISPR and analysis of variance (ANOVA) systematic analyses. Fusions events are annotated if one of the partner genes is significantly overexpressed using our linear regression model, contain a COSMIC cancer driver gene, or has been detected in patient samples

pave the way for their clinical application. More broadly, these results support the use of validated oncogenic fusions as therapeutic biomarkers in diverse histologies, and the utility of basket trials for clinical development of drugs targeting fusion proteins irrespective of tumour type, such as the type used for the development of entrectinib in solid tumours with *ALK*, *ROS1*, and *NTRK* fusions[50]. A notable exception in our analysis was the differential sensitivity to WNT pathway inhibition of CRC vs. biliary tract and esophageal cancer cell lines with canonical R-spondin fusions. This suggests that tissue context could impact the functional role of some fusions as has been observed for oncogenes (e.g., *BRAF*-mutated CRC[51]). Further investigations are warranted to understand this difference, and drug combinations could be evaluated in these specific context to overcome resistance similar to what is in clinical development for *BRAF*-mutated CRC[51].

We identified and functionally evaluated less well-studied gene fusions, as exemplified by *YAP1-MAML2* rearrangements, which are required for cell fitness in multiple histology and associated with increased YAP1 signaling. Given the emerging role of YAP1/TEAD1 and the Hippo pathway in cancer, there is interest in pharmacological inhibition of Hippo signaling as an anti-cancer therapeutic strategy[52]. We provide preclinical evidence supporting inhibition of this signaling axis in *YAP1-MAML2*-fusion-positive tumours, which could pave the way for clinical development in a rare but defined patient population.

In summary, our findings that most fusions tested were dispensable for tumour cell fitness has implication for the interpretation of fusions detected in tumour sequencing data. Furthermore, this observation supports the use of systematic functional studies in preclinical models as an unbiased platform to systematically assess the impact of fusions in cancer. Extending this approach to a larger

set of cancer models that represents the histopathologic and genomic diversity of patient tumours could reveal additional new insights with clinical relevance. Notably, here we identified fusion drivers of carcinogenesis, which could represent targets for drug development and specific actionable leads with potential for immediate clinical development in defined fusion-positive patients.

## Methods

**Sample selection**. A collection of 1011 cancer cell lines have been compiled from publicly available repositories as well as private collections and maintained following supplier guidelines. These cell lines were selected to be genetically unique based on short tandem repeat (STR) and single-nucleotide polymorphism (SNP) fingerprints (http://cancer.sanger.ac.uk/cell_lines/download). STR profiles matched those in public repositories or match published STR profiles. The cells lines have been extensively characterized using whole-exome sequencing (EGAD00001001039), Affymetrix SNP6-based copy number analysis, and genotyping (EGAD00010000644). The origin of all cell lines and related data are available through the Cell Model Passports database (https://cellmodelpassports.sanger.ac.uk/)[9]. All cell lines have been tested for mycoplasma using a PCR-based assay (EZ-PCR Mycoplasma Detection Kit, Biological Industries) and a biochemical test (MycoAlert, Lonza). Cell lines that test positive using either method are removed from the collection. Comparing SNP6 genotyping data and somatic variant data, we verified that cell lines overlap extensively with those characterized by the Cancer Cell Line Encyclopedia (TCGA). See Supplementary Data 1 for a complete description of the cell lines and their molecular annotation. This study includes the following commonly misidentified lines: TE-12, NCI-H1304, MEL-HO, U-118-MG, BT-549, BE-13, ETK-1, GT3TKB, MKN28, RPMI-6666, SK-MG-1, CGTH-W-1, H513, OVMIU, and KP-1N. Misidentified lines have been noted in Supplementary Data 1 and on the Cell Model Passport (https://cellmodelpassports.sanger.ac.uk). Mis-identification does not impact tissue of origin, genomic data used for analyses, or results. SNU1411 were purchased from KCLB and maintained in RPMI-1640 with L-glutamine (300 mg/L), 25 mM HEPES and 25 mM NaHCO₃, 90%; heat-inactivated fetal bovine serum, 10%.

**RNA-seq data and identification of fusion transcripts**. RNA-seq data for 589 cell lines was obtained from the CGHub, and 450 cell lines were sequenced at the Sanger Institute (EGAS00001000828). For 23 cell lines, sequence was obtained from both CGHub and Sanger Institute to allow comparison of the output based on the sequence from the two studies. Where replicated datasets were available, we took forward only fusions called from Sanger Institute sequencing data for our final analysis.

For sequencing performed at the Sanger Institute, cell line pellets were collected during exponential growth in RPMI or Dulbecco's modified Eagle's medium/F12 and were lysed with TRIzol (Life Technologies) and stored at −70 °C. Following chloroform extraction, total RNA was isolated using the RNeasy Mini Kit (Qiagen). DNAse digestion was followed by the RNAClean Kit (Agencourt Bioscience). RNA integrity was confirmed on a Bioanalyzer 2100 (Agilent Technologies) prior to labeling using 3′ IVT Express (Affymetrix). Sequence libraries were prepared in an automated fashion on the Agilent Bravo platform using the stranded mRNA Library Prep Kit from KAPA Biosystems. Processing steps were unchanged from those specified in the KAPA manual, except for use of an in-house indexing set. Three publicly available gene fusion detection algorithms were used (TopHat-Fusion (v2.1.0), STAR-Fusion (v2.5.0), and deFuse (v0.7.0)) as described in GitHub (https://github.com/cancerit/cgpRna/blob/dev/README.md).

**Fusion transcript filtering criteria**. From the output of the three distinct fusion detection algorithms, we selected for analysis only fusions that were called with four or more reads that align directly across the breakpoint. We also required fusions to be called by at least two different algorithms. Next, we removed fusions identified from the analysis of 245 non-neoplastic samples downloaded from GTEx[53].

**Fusion annotations**. The frame of fusion transcripts was predicted using the GRASS algorithm that is built into the fusion-calling pipeline (https://github.com/cancerit/cgpRna/blob/dev/README.md/https://github.com/cancerit/grass). A list of known cancer driver genes was obtained from the COSMIC cancer census in June 2017 (https://cancer.sanger.ac.uk/cosmic/curation). Known cancer fusions, as well as annotation of TSGs and oncogenes, were downloaded from the COSMIC fusion census (https://cancer.sanger.ac.uk/cosmic/fusion). Our reference set of fusions identified previously in patient samples comes from an analysis of fusions in over 9000 TCGA samples[15].

**PCR validation of fusion transcript**. Complementary DNA (cDNA) was prepared using the Superscript double-stranded cDNA Synthesis Kit (Invitrogen) followed by SPRI (Solid Phase Reversible Immobilization) clean-up. The cDNA was then subjected to "whole-genome amplification" using the Illustra GenomiPhi HY DNA Amplification Kit as per the manufacturer's instructions. This WGA'ed cDNA was used as a template for the PCR validation. Generally two distinct sets of PCR primers were designed using Primer3 (http://www.bioinformatics.nl/cgi-bin/

primer3plus/primer3plus.cgi) for each fusion junction tested (Primer sequences are in Supplementary Data 9). The primers were then checked by ePCR (http://www.ncbi.nlm.nih.gov/sutils/e-pcr/reverse.cgi?taxid=9606&db=2&orgdb=118&margin=200&mism=0&gaps=0) against the genome and transcriptome to make sure that they would not produce a PCR product of <5 kb. PCRs were carried out in duplicate using two PCR programs: (1) 30 cycles, 95 °C for 30 s, 60 °C for 30 s, and 72 °C for 30 s; (2) a touchdown program reducing annealing temperature by 2 °C every two cycles, dropping from 60 °C to 50 °C over ten cycles, with a final 20 cycles at 50 °C. For all cycles, the melting and extension temperatures were 95 °C and 72 °C, respectively, and all stages were maintained for 30 s. Finally, the PCRs were checked by gel electrophoresis to confirm the presence of a product of the predicted size. To validate candidate fusions, PCR products were further checked by PCR product Sanger sequencing. In these cases, the PCR products were first cleaned up using ExoSAP (Affymetrix) and then capillary sequenced by Eurofins Genomics (Ebersberg, Germany).

**Differential fusion frequencies across cancer types**. We designed an empirical permutation approach to identify genes with differential fusion frequencies across cancer types. Briefly, we built a binary matrix (genes × samples), where 1 indicates that the sample contains at least one fusion involving a gene $G$ and 0 indicates that the gene $G$ is not fused. Under the null hypothesis that gene alterations distribute homogeneously across cancer types (i.e., any sample from any cancer type has the same likelihood of having the gene $G$ fused), we permuted 10,000 times the initially observed matrix using the algorithm BiRewire implemented in an R package[54]. This algorithm generates randomized binary networks that preserve marginal totals. Randomized networks are used to build an empirical null distribution, modeling the likelihood of observing a gene fused in $X$ samples. Thus, we can derive a $p$ value per gene and cancer type representing the probability of observing ≥$N$ samples with the gene $G$ fused in the null distribution. Nominal $p$ values are adjusted using the FDR method.

**Gene expression and differential gene expression analysis**. Read counts per gene, based on the union of all exons from all possible transcripts, were used to calculate reads per kilobase per million (RPKM) as described previously[10]. To identify genes which expression is significantly altered when fused, we used MLR. For each fusion-associated gene, the expression values $G$ ($\log_2$ RPKM) from each sample $S$ are modeled as a function of the fusion status of the gene in the sample $S$ ($X_{fusion}$), a series of dependent covariates ($X_{covariates}$, including cancer type in pan-cancer analyses, microsatellite instability status, and gene absolute CNA status), and a noise term ($\psi$):

$$G = \beta_{covariates}X_{covariates} + \beta_{fusion}X_{fusion} + \psi. \qquad (1)$$

The association between the fusion status and gene expression was defined by the regression coefficient ($\beta_{fusion}$) estimated with a multiple linear least-squares regression. Significance of the regressors was estimated with a type II ANOVA method from the car R package. For each cancer type, $p$ values were adjusted for multiple testing correction using the Benjamini–Hochberg method. To annotate fusion genes for overexpression of the 3′ gene, we selected fusions where the expression level of the 3′ gene is above the 95th percentile (i.e., 95% of the cell lines have an expression level lower than the one observed in the cell line carrying the fusion). Z-score RNA-seq values for TCGA samples from esophageal adenocarcinoma (187 samples) and lung squamous cell carcinoma (187 samples) were downloaded from cBioPortal[55].

**Gene-set enrichment analysis**. Data for GSEA and expression of neuroendocrine markers for GSEA analysis, and RNA-seq voom-transformed gene expression measurements[56] were obtained from ref. [10]. GSEA software was downloaded from the Broad Institute GSEA portal (http://software.broadinstitute.org/gsea/index.jsp) and was applied using default parameters and exploiting signal-to-noise metric for gene ranking. The significance of enrichment was estimated using 1000 gene permutations. Heatmap of neuroendocrine genes in small-cell lung cancer cell lines was generated by the GEDAS software 1.1.6 Beta[57].

**Cancer functional event–drug association analysis**. Potential confounding factors in our fusion–drug association analysis were identified by implementing a systematic analysis of associations between cancer functional events published by Iorio et al. [8] for our panel of cell lines and our set of 409 drugs. The 717 cancer functional events used in the analysis included 281 genes with somatic coding mutations, 424 copy number altered chromosomal segments and methylation status for 12 segments that included any gene altered by point mutations. The analysis was conducted as described by Garnett et al. [7], although our analysis only considered tissue type and microsatellite instability status as covariates.

We identified 101 large-effect size significant cancer functional event–drug associations across 73 drugs when implementing the same cut-offs as in ref. [8] (FDR <25%, $p$ value <0.001, and positive and negative Glass Deltas >1) reported in Supplementary Data 6.

**Gene fusion–drug association analysis**. The fusion ANOVA model was constructed as per Garnett et al. [7], but also includes as covariate any cancer functional event that was involved in a significant large-effect size association with a given

drug. High-throughput drug sensitivity data were generated by the GDSC project (www.cancerRxgene.org) at the Sanger Institute as previously described[8]. Details of compounds screened and cell line sensitivity data are provided in Supplementary Data 5.

**CRISPR screening data analysis**. sgRNA raw counts and guide annotations were obtained for 274 cell lines screened as part of Project Achilles by the Broad Institute[24] (Supplementary Data 10), 14 acute myeloid cell lines screened by Wang et al. [6] (Supplementary Data 11), and 206 cell lines from Project Score[23] performed at the Sanger Institute (Supplementary Data 12). CRISPR screening data for EGI-1, ESO51, H3118, SAS, SBC-3, and SNU1411 were generated ad hoc for this study at the Sanger Institute, following the CRISPR screening pipeline used to screen cell lines part of the Project Score. sgRNA target positions were converted from GrCh37 to GrCh38 using the NCBI remapper tool (https://www.ncbi.nlm.nih.gov/genome/tools/remap).

sgRNA raw counts were converted into log fold changes and corrected for CRISPR biases using CRISPRcleanR[58] (https://github.com/francescojm/CRISPRcleanR). Corrected log fold changes were scaled to essentials using the scale-to-essentials R function in the CERES package (https://github.com/cancerdatasci/ceres/blob/master/R/scale_to_essentials.R).

Positions of sgRNA were mapped onto fusion transcripts using a bespoke R script that takes into consideration: (1) the fusion transcript breakpoint and (2) the mapping location of the guides. Essentially, sgRNA map onto the 5′ gene if the gene is on the positive DNA strand and the guides maps before the position of the fusion breakpoint, or if the gene is on the negative DNA strand and the sgRNA maps after fusion breakpoint. Guides map onto the 3′ gene if the opposite is true.

In order to calculate an FES, we took the following steps: (1) Z-normalized the scaled and corrected log fold changes across all cell lines screened for a given sgRNA; (2) subtracted the mean Z-score of non-mapping from that of the mapping sgRNA for both genes involved in a fusion transcript. Where a gene has no non-mapping sgRNA, the difference was taken from zero; (3) values obtained in step 2 for each gene were averaged to produce the FES.

To assign statistical significance for the FES, we performed 10,000 randomizations of all scaled and corrected log fold changes within cell lines. Randomized FES were then calculated for each fusion transcript, as described in the above paragraph. A $p$ value of statistical significance was assigned to each fusion transcript as the fraction of randomized FES that have a higher FES score than the non-randomized FES. For multiple-hypothesis correction, we calculated a false discovery rate for each $p$ value. Calculation of the FES and the subsequent randomization was performed independently for each data resource, since the sgRNA libraries were different in all cases. Based on an empirical analysis of "gold standard" known functional fusions, we removed any significant hits where the difference between the average scaled fold changes of mapping and non-mapping guides is <0.45.

**Cell viability assay**. Cell lines were seeded at different densities ($1–3 \times 10^3$ cells per well) in 100 μl complete growth medium in 96-well plastic culture plates at day 0. The following day, serial dilutions of drug were added to the cells in an additional 50 μl of medium. Plates were incubated at 37 °C in 5% $CO_2$ for 5 days, after which the cell viability was assessed by measuring ATP content through Cell Titer-Glo Luminescent Cell Viability assay (Promega). Luminescence was measured by Envision Multiplate Reader at day 7. Crystal violet growth assays were performed seeding $30–50 \times 10^3$ cells in 6-well plates. After 24 h, the medium was replaced adding drugs as indicated. After 7–10 days of treatments, cells were fixed with a solution of 3% paraformaldehyde and then stained with 0.05% crystal violet in distilled water.

**PDX database**. Gene fusion data for 126 pancreatic adenocarcinoma PDX models were downloaded from HuBase database (https://hubase.crownbio.com/; Crown Bioscience International, Santa Clara, CA, USA).

**Interphase and fiber FISH**. Metaphase chromosomes were prepared from cell lines listed in Supplementary Data 10 using a standard method. Briefly, colcemid (Thermo Fisher Scientific) was added to a final concentration of 0.1 mg/ml for 1 h, followed by treatment with hypotonic buffer (0.4% KCl in 10 mM HEPES, pH 7.4) for 10 min and subsequent fixation using 3:1 (v/v) methanol–acetic acid. Human fosmid and bacterial artificial chromosome (BAC) clones containing the genes of interest (Supplementary Table 1) list of BAC and fosmid clones used in the FISH validation) were provided by the clone archive team of the Wellcome Sanger Institute. Probes were generated by whole-genome amplification with GenomePlex Whole Genome Amplification Kits (Sigma-Aldrich), from purified BAC fosmid DNA as described previously[59]. For interphase and metaphase FISH, probes were labeled directly with Atto488-XX-dUTP, Cy3-XX-dUTP, Texas Red-12-dUTP, and Cy5-XX-dUTP (Jena Bioscience), respectively. Slides pre-treatment included a 10 min fixation in acetone (Sigma-Aldrich), followed by baking at 65 °C for 1 h. Metaphase spreads on slides were denatured by immersion in an alkaline denaturation solution (0.5 M NaOH, 1.0 M NaCl) for 7–10 min, followed by rinsing in 1M Tris-HCl (pH 7.4) solution for 3 min, 1× PBS for 3 min and dehydration through a 70, 90, and 100% ethanol series. The probe mix was denatured at 65 °C for 10 min before being applied onto the denatured slides. Hybridization was performed at 37 °C overnight. The post-hybridization washes included a 5 min stringent wash in 1× SSC at 73–75 °C, followed by a 5 min rinse in 2×

SSC containing 0.05% Tween®20 (VWR) and a 2 min rinse in 1× PBS, both at room temperature. Finally, slides were mounted with SlowFade Gold® mounting solution containing 4′,6-diamidino-2-phenylindole (DAPI) (Thermo Fisher Scientific). Indirectly labeled probes were used in fiber FISH with single-molecule DNA fibers. Single-molecule DNA fibers for fiber-FISH were prepared by molecular combing. The three-color probe set was labeled with biotin-16-dUTP, DNP-11-dUTP, and digoxenin-11-dUTP (Jena Bioscience), respectively, and visualized with Cy3-conjugated, FITC-conjugated, and Texas Red-conjugated antibodies, with the exception of post-hybridization washes, which consisted of three 5 min washes in 2× SSC at 42 °C, instead of two 20 min washes in 50% formamide/50% 2× SSC at room temperature. Slides were examined using AxioImager D1 microscope equipped with appropriate narrow-band pass filters for DAPI, Aqua, FITC, Cy3, Texas Red, and Cy5 fluorescence. Digital images capture and processing were carried out using the SmartCapture software (Digital Scientific UK). Ten randomly selected metaphase cells were karyotyped based on the multiplex FISH and DAPI-banding patterns using the Smart-Type Karyotyper software (Digital Scientific UK).

## Data availability

All data are available in the main text or the Supplementary information. Supplementary Data 10, 11, and 12 are accessible at the following link https://figshare.com/s/169e9fa07450ea7cebce.

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

## Acknowledgements

We thank the Garnett laboratory, Cellular Genetics and Phenotyping facility, and drug screening teams at the Sanger Institute for data generation and assistance. We also thank Sandra Louzada for support with DNA probe preparation. We would also like to thank Serena Nik-Zainal and Andrea Degasperi for useful discussions.

## Author contributions

The project was conceived by M.J.G., G.P., E.D.C., and U.M. Molecular biology experiments were designed by G.P. and E.D.C., and performed by G.P. Computational analyses were designed by E.D.C. and G.P. and performed by E.D.C. RNA-seq data pre-processing was performed by G.B. and A.M. Drug screening data generation was led by M.J.G., C.H.B., and U.M. Gene expression analysis was performed by L.G.A. GSEA was performed by G.P. CRISPR-Cas9 screening data was provided by F.M.B., F.I., E.G., E.S., K.Y. and M.J.G. Statistical and computational advice was provided by E.G. and F.I. PCR validations was performed by E.A. FISH experiments were performed by B.F., R.B., and F.Y. Project supervision was assisted by A.B., D.D., J.S-R., and E.S. The manuscript was written by M.J.G., G.P., and E.D.C. All authors edited and approved the manuscript. Funding was acquired by M.J.G. and J.S-R. The project was administered by M.J.G.

## Additional information

**Competing interests:** E.S. and D.D are employees of GSK. U.M. is an employee of AstraZeneca. M.J.G receives funding from AstraZeneca and performed consultancy for Sanofi. This works was funded by Open Targets which is a public-private initiative involving industry and academia. The remaining authors declare no competing interests.

