## [Peer Review File · Nature Communications]

Reviewers' comments:

Reviewer #1 (Remarks to the Author):

Picco, Garnett and colleagues report large-scale functional annotation of fusion genes across cancer types. They analysed >1000 cell lines and used a CRISPR-based approach to interrogate and validate the role of thousands of fusions for cancer cell fitness. Moreover, they performed high-throughput drug screens to identify therapeutically actionable fusions in different entities. The study reports numerous novel discoveries and validates a series of screening results.

This is an important study from a leading group in the field. It is to my knowledge the first comprehensive large-scale survey on the functional role of gene fusions. The study produces large data sets and numerous new discoveries, of which some have potential translational relevance. I only have few minor comments:

Please discuss limitations of the CRISPR-based approach to validate fusions, e.g. point out clearly what percentage of fusions were testable using the available CRISPR-screening data and gRNA libraries. Please discuss also methodological and technical limitations (estimated percentage of false-negative results etc).

BRD4-NUTM1 fusions in small cell lung cancer. It seems that the genetics of the SBC-3 cell line is not reminiscent of SCLC. Is it possible that the cell line was misclassified when it was generated decades ago? More information would be beneficial, e.g. about the history of the cell line, age of the patient, transcriptional profiles (clustering with SCLC or NUT midline carcinoma?).

Reviewer #2 (Remarks to the Author):

This is a very interesting study that has taken a rigorous, systematic approach to mine chimeric fusion genes in cancer cell lines and assess functionality and targetability, particularly using genome editing approaches that are nicely designed to distinguish native from fusion transcript. The authors have provided a really useful resource and multiple (largely anecdotal) insights into the role of chimeric fusions.

There are several key points that the authors should be able to address.

As a cell line study, many cases and subtypes of key tumor types are not represented. For example, from this referees perspective, the majority of important acute leukemia subtypes lack representation in cell lines, but most are most definitely driven by gene fusions. Thus the statement that most fusions are not functional needs to be qualified.

The manuscript, I think, is focused on rearrangements that results in in frame chimeric fusion genes. This needs to be better explained - have the analyses examined and distinguished between non-in frame chimeras (E.g. chimeras that result in loss of function of the downstream gene) and enhancer hijacking events that can also be identified by RNA-seq analysis? This is very important and must be addressed. In frame chimeras are only one consequence of gene rearrangements.

A vexed issue in the field is the ability of RNA-seq to predict clonality of fusions. This is a central point of the manuscript - ie if one takes all clonal fusions, how does that influence driver/druggability status? The authors should provide a clearer workflow and results for how clonality assays were prioritized, performed, and if there are differences in cell dependencies between clonal and non clonal events.

The authors, appropriately, use multiple fusion calling algorithms, but the numbers of events

called by each is very large. The approach of using an intersected N of calls as likely events is not satisfying as a bad caller can throw the whole analysis out. The authors need to systematically define the accuracy of each caller and define criteria that more robustly call a fusion as real. This requires wet lab validation.

p4 - multiple transcripts in the same cell line - please clarify. Same partner genes but different exon-exon break points, or other?

We thank the reviewers for their constructive comments on our manuscript **“Functional linkage of gene fusions to cancer cell fitness assessed by pharmacological and CRISPR/Cas9 screening”**. We provide below a point by point response to their comments and modified the manuscript to reflect these changes. The original comments are italicized and our response are in bold text. To aid the reviewers, where the manuscript has been modified to reflect their comments the text is in red font.

Reviewer 1

Picco, Garnett and colleagues report large-scale functional annotation of fusion genes across cancer types. They analysed >1000 cell lines and used a CRISPR-based approach to interrogate and validate the role of thousands of fusions for cancer cell fitness. Moreover, they performed high-throughput drug screens to identify therapeutically actionable fusions in different entities. The study reports numerous novel discoveries and validates a series of screening results.

This is an important study from a leading group in the field. It is to my knowledge the first comprehensive large-scale survey on the functional role of gene fusions. The study produces large data sets and numerous new discoveries, of which some have potential translational relevance. I only have few minor comments:

Please discuss limitations of the CRISPR-based approach do validate fusions, e.g. point out clearly what percentage of fusions were testable using the available CRISPR-screening data and gRNA libraries. Please discuss also methodological and technical limitations (estimated percentage of false-negative results etc).

We thank the reviewer for their positive feedback and commenting on the value of our study.

Using our CRISPR-based approach the total number of tested fusion transcripts is 2,821 (26% of all possible) and this is now clearly stated in the manuscript. To discuss the limitations of our approach we have included the following paragraph in the discussion:

“Although our study suggests that most fusions are not functional, there are several limitations to our approach. Some fusions may be required for aspects of the malignant phenotype not measured here, such as tumor initiation, paracrine signaling, host–tumor cell interaction and metastasis. Our CRISPR-based approach is only suitable for testing fusions with mapping sgRNA, and is subject to possible bias based on sgRNA efficiency and the number of mapping versus non-mapping sgRNA. Furthermore, this approach captures fusions which induce gain-of-function or dominant-negative effects, but is not able to identify recessive effects such as inactivation of a tumour suppressor. Some tumour types are relatively poorly represented, including haematological and pediatric malignancies, and it may be that

in some tumour types we observe different patterns of functionality. Finally, sub-clonal fusions could lead to false negative results. Despite these limitations, our finding that most fusions tested do not have supporting functional evidence, including fusions with cancer drivers genes, emphasises the importance of analyses to ascribe function when interpreting fusions identified using genomic sequencing.”

BRD4-NUTM1 fusions in small cell lung cancer. It seems that the genetics of the SBC-3 cell line is not reminiscent of SCLC. Is it possible that the cell line was misclassified when it was generated decades ago? More information would be beneficial, e.g. about the history of the cell line, age of the patient, transcriptional profiles (clustering with SCLC or NUT midline carcinoma?).

We agree that the **BRD4-NUTM1** fusion in the SBC cell line is an intriguing case and have investigated this further to determine if it could be a mis-classification. SBC-3 was established from a 24 year old male with SCLC¹ which is most consistent with the median age of NUT midline carcinoma (median age = 16 - 22 years) compared to SCLC (median age = 71 years). NMCs of the lung have been reported, but they are extremely rare². In support of a role for NUTM1 fusions in different tissue histology, we note that recent studies established that **NUT-associated fusions** can occur in tumors outside the midline, such as soft tissue, brain, and kidney³.

Furthermore, we analyzed transcriptional data by performing unsupervised clustering. We found that SBC-3 clustered with RPMI2650, a head and neck cancer cell line of NMC origins, reinforcing that a common pathogenomic alteration can induce similar gene expression. These two samples cluster together with a small set of SCLC cell lines, quite far away from the majority of SCLC and lung cancer cell lines. Notably, RPMI2650 does not cluster with head and neck cancer cell lines. As mentioned in the manuscript and in supplementary Figure 4, SBC3 do not display genetic alterations (absence of **RB1** or **TP53** mutations) and neuroendocrine marker typical of SCLC, nevertheless, a subset of other SCLC cell lines (n>4) display a similar de-differentiated status, making this an imperfect marker of differential diagnosis. Mining TCGA data we found a lung squamous cell carcinoma derived from a 39-years old patient, positive for NSD3-NUTM1 fusion and characterized by an aberrant overexpression of the NUTM1 transcript.

Overall, we are not able to exclude the possibility that SBC-3 is a misclassified NMC of the lung. Nonetheless, our results suggest that a subset of tumours which are diagnosed as lung cancer possess a potentially clinically actionable NUTM1 fusion.

We have revised the manuscript to discuss the possibility of misclassification, and we now make the distinction that NUTM1 fusions have been reported in tumours diagnosed as lung cancer, rather than stating that there are a subset of lung cancers which have NUTM1 fusion.

Reviewer 2

This is a very interesting study that has taken a rigorous, systematic approach to mine chimeric fusion genes in cancer cell lines and assess functionality and targetability, particularly using genome editing approaches that are nicely designed to distinguish native from fusion transcript.

The authors have provided a really useful resource and multiple (largely anecdotal) insights into the role of chimeric fusions.

We thank the reviewer for their positive comments.

There are several key points that the authors should be able to address.

As a cell line study, many cases and subtypes of key tumor types are not represented. For example, from this referees perspective, the majority of important acute leukemia subtypes lack representation in cell lines, but most are most definitely driven by gene fusions. Thus the statement that most fusions are not functional needs to be qualified.

We thank the reviewer for making this important distinction and we now state in the manuscript that some tumour types are poorly represented in our dataset, and these may have different patterns of functionality:

“Finally, some tumour types are poorly represented in the CRISPR data, including haematological and pediatric malignancies, and it may be that in some tumour types we observe different patterns of functionality.”

We note that even in acute leukemia subtypes, which are known to be driven by gene fusions, it is likely that there will be passenger fusion transcripts reported. Thus, our statement that there is a lack of supporting functional evidence for most tested fusions is accurate. To make this important point clear, we now include in the manuscript the percentage of tested fusion with functional evidence, as well as the percentage of tested cell lines for which we identify at least one functional fusion.

The manuscript, I think, is focused on rearrangements that results in in frame chimeric fusion genes. This needs to be better explained - have the analyses examined and distinguished between non-in frame chimeras (E.g. chimeras that result in loss of function of the downstream gene) and enhancer hijacking events that can also be identified by RNA-seq analysis? This is very important and must be addressed. In frame chimeras are only one consequence of gene rearrangements.

We have included all fusions in our systematic analysis of fusions, including in frame and non-in frame fusions transcripts. We believe this comprehensive approach gives the most global view of fusions. Nonetheless, we thank the reviewer for raising this important distinction between in frame fusions that generate chimeric proteins, and fusions which induce enhancer hijacking and dominant negative effects. To make

this distinction clear, we now clearly state all fusions are considered in our systematic approach. We also include a sentence in the discussion to make clear that our CRISPR-based approach is not powered to identify loss-of-function fusion drivers. Furthermore, we distinguish between in frame and non-in frame fusions in our analysis for enrichment of functional fusions in different gene sets (Fig. 4b)

A vexed issue in the field is the ability of RNA-seq to predict clonality of fusions. This is a central point of the manuscript - ie if one takes all clonal fusions, how does that influence driver/druggability status? The authors should provide a clearer workflow and results for how clonality assays were prioritized, performed, and if there are differences in cell dependencies between clonal and non clonal events.

Fusion clonality is a good, outstanding question, and something we have considered. Readouts of clonality from FISH analyses carried out as part of our validation strategy demonstrated that ATG7-RAF1 and NUTM1-BRD4 fusions were detected in >90% of cell nuclei analyzed by FISH. So far, all the fusion genes that we assayed for validation purposes were clonal. We now include these data in the manuscript.

Beyond the work described above, ideally we would have liked to include a systematic approach to address this questions across our large dataset. However, to the best of our knowledge, there are no standard protocols in the community that are able to evaluate fusion clonality from RNA-Seq data reliably in a high-throughput manner. We are aware of the work by Morgan et al (2018) ⁴ examining the clonality of around 40 fusions using droplet digital PCR that uses custom probes designed on the fusion breakpoint. The number of fusions in this work is relatively low, as only 265 commonly rearranged genes are tested using a targeted sequencing panel. However, implementing this on our ~10,000 fusion transcripts is prohibitive.

We are also aware of the work by Gu et al (2017) ⁵ that examines all RNA sequencing reads around the proposed transcript breakpoint and calculates a ratio (iFCR) of fused reads vs non-fused parent gene reads to estimate relative clonal size in a tumour. This paper is a valuable effort at estimating the clonality of fusions using a computational approach based on existing data. However, the iFCR does not take into account copy number of the genes involved in the fusion, which could confound the calculation of the iFCR if either the fusion or parent genes are situated within a region of differential copy number. This is particularly problematic across such a large and diverse panel of cancer cell lines, some of which are largely driven by copy number alterations (e.g. breast and ovarian) whereas others are driven by mutations. This could introduce a strong bias in our analysis. In addition, although the iFCR ratio provides an estimate of relative heterogeneity, it is unclear how this relates to the absolute clonality of a fusion in sample - ie. not able to provide % of cells. Thus, based on these considerations, and that no 'gold standard' approach has been described to calculate clonality from RNAseq data, we believe that this question, while valuable, is beyond the scope of our project. However, we believe that the reviewer is right in raising that clonality is an important question to address in the future and

have updated the main text to reflect that clonality is a potential limitation of our approach.

“Some tumour types are relatively poorly represented in the CRISPR data, including haematological and pediatric malignancies, and it may be that in some tumour types we observe different patterns of functionality. Finally, sub-clonal fusions could lead to false negative results. “

The authors, appropriately, use multiple fusion calling algorithms, but the numbers of events called by each is very large. The approach of using an intersected N of calls as likely events is not satisfying as a bad caller can throw the whole analysis out. The authors need to systematically define the accuracy of each caller and define criteria that more robustly call a fusion as real. This requires wet lab validation.

Given the recognised difficulty in accurately calling fusion transcripts, the use of multiple calling algorithms is becoming the standard approach ⁶. Indeed, for our analysis, we required a fusion to be called by 2 of 3 fusion calling algorithms. We adopted this approach because it is unlikely to incorporate a large number of false positive calls detected by a single algorithm, nor contribute to false negatives due to a fusion being missed by one algorithm. We believe this is a reasonable approach to empower novel discovery while minimising as much as possible both false positive and negative results introduced by a single caller.

In addition, as suggested, we have taken a number of steps to rigorously define the accuracy of our approach and validate specific fusions. These include:

1. We performed 925 PCR reactions across 23 cell lines to determine the performance of each of the fusion calling algorithms. Validation rates for DeFuse, TopHat-fusion and STAR were 56%, 34% and 14%, respectively. When applying a threshold of calling by two algorithms, we observe an average PCR validation rate of 63%. These percentages are impacted by the thresholds and settings used for each calling algorithm and should not be taken as a hard guideline. They do however show, that each algorithm does identify validating fusions and can therefore positively contribute to the final selection. Of note, when considering only the subset of 10,514 fusion transcripts passing our filtering criteria, 72% of fusions (406 of 535 fusions) tested were validated using our PCR based approach. This is reported in the manuscript. Previous papers that analyse gene fusions from RNA-Seq data report similar validation rates of about 69%⁶.
2. We assessed the likelihood of false positives using 23 cancer cell lines for which RNA-Seq data was available from both CGHub, as well as via in-house sequencing. There, we assessed the proportion of fusion transcripts called in the same cell line in both data sets. We found that despite relatively high rates of PCR validation, this proportion was low for individual algorithms (3-30%), and could be significantly improved by only considering fusion transcripts that are called by at least two different fusion calling algorithms (70%).

3. Each of the specific fusions discussed in detail in the manuscripts, including fusions of RAF1, BRD4, NUTM1 and YAP, have been validated by both PCR and FISH.
4. We find that the frequency of fusions per cancer type is similar to what has been reported in patient tumours, suggesting that we do not report a large number of false positives. Conversely, we identified known oncogenic fusions in our panel of cell lines, for example EWSR1-FLI1 in Ewing's, NPM1-ALK fusions in lung cancer, TMPRSS2-ERG and BCR-ABL1, specifically in the tissues/histology they are observed clinically.

Collectively, we believe these analyses demonstrate the accuracy of our approach and provide confidence in the fusions identified.

p4 - multiple transcripts in the same cell line - please clarify. Same partner genes but different exon-exon break points, or other?

We have now clarified in the text that multiple transcripts for the same fusion in the same cell lines represent different breakpoints between the same gene partners:

“The presence of a fusion in a cell line, even in instances where multiple transcripts involving the same partner genes were detected, was defined as a ‘fusion event’. Thus, we identified 8,354 gene fusion events (from 10,514 fusion transcripts) and, because only a small number of fusions were recurrent, a total of 7,430 unique fusions (Supplementary Table 2).”

References:

1. Miyamoto, H. Establishment and characterization of an adriamycin-resistant subline of human small cell lung cancer cells. *Acta Med. Okayama* **40**, 65–73 (1986).
2. Sholl, L. M. *et al.* Primary Pulmonary NUT Midline Carcinoma: Clinical, Radiographic, and Pathologic Characterizations. *J. Thorac. Oncol.* **10**, 951–959 (2015).
3. Schaefer, I.-M. *et al.* CIC-NUTM1 fusion: A case which expands the spectrum of NUT-rearranged epithelioid malignancies. *Genes Chromosomes Cancer* **57**, 446–451 (2018).
4. Morgan, G. J. *et al.* Kinase domain activation through gene rearrangement in multiple myeloma. *Leukemia* **32**, 2435–2444 (2018).

5. Gu, J.-L. *et al.* RNA-seq Based Transcription Characterization of Fusion Breakpoints as a Potential Estimator for Its Oncogenic Potential. *Biomed Res. Int.* **2017**, 9829175 (2017).
6. Gao, Q. *et al.* Driver Fusions and Their Implications in the Development and Treatment of Human Cancers. *Cell Rep.* **23**, 227–238.e3 (2018).
4. Gu, J.-L. *et al.* RNA-seq Based Transcription Characterization of Fusion Breakpoints as a Potential Estimator for Its Oncogenic Potential. *Biomed Res. Int.* **2017**, 9829175 (2017).

REVIEWERS' COMMENTS:

Reviewer #1 (Remarks to the Author):

The authors addressed all my comments. I recommend publication.

Reviewer #2 (Remarks to the Author):

The authors have throughfully responded to all my concerns. It is an interesting and impressive study. I remain surprised about the low concordance between the fusion callers but the authors have been open about this and have used the intersection of two callers to priortize fusion calls for further analysis.

[Charles Mullighan]